# Effect of a Park-Based Physical Activity Intervention on Psychological Wellbeing at the Time of COVID-19

**DOI:** 10.3390/ijerph19106028

**Published:** 2022-05-16

**Authors:** Stefania Toselli, Laura Bragonzoni, Alessia Grigoletto, Alice Masini, Sofia Marini, Giuseppe Barone, Erika Pinelli, Raffaele Zinno, Mario Mauro, Pietro Loro Pilone, Sonia Arduini, Simona Galli, Mauro Vitiello, Bruno Vicentini, Giorgia Boldrini, Muriel Assunta Musti, Paolo Pandolfi, Maurizio Liberti, Gerardo Astorino, Pasqualino Maietta Latessa, Laura Dallolio

**Affiliations:** 1Department of Biomedical and Neuromotor Sciences, University of Bologna, 40126 Bologna, Italy; stefania.toselli@unibo.it (S.T.); alessia.grigoletto2@unibo.it (A.G.); laura.dallolio@unibo.it (L.D.); 2Department of Life Quality Studies, University of Bologna, 47921 Rimini, Italy; laura.bragonzoni4@unibo.it (L.B.); sofia.marini2@unibo.it (S.M.); giuseppe.barone8@unibo.it (G.B.); erika.pinelli2@unibo.it (E.P.); raffaele.zinno2@unibo.it (R.Z.); pasqualino.maietta@unibo.it (P.M.L.); 3Department of Public Health, AUSL di Bologna, 40124 Bologna, Italy; p.loropilone@ausl.bologna.it (P.L.P.); sonia.arduini@ausl.bologna.it (S.A.); galli.simona1991@gmail.com (S.G.); mauro.vitiello@ausl.bologna.it (M.V.); murielassunta.musti@ausl.bologna.it (M.A.M.); paolo.pandolfi@ausl.bologna.it (P.P.); maurizio.liberti@ausl.bologna.it (M.L.); gerardo.astorino@ausl.bologna.it (G.A.); 4Culture and City Promotion Department, Municipality of Bologna, 40126 Bologna, Italy; bruno.vicentini@comune.bologna.it (B.V.); giorgia.boldrini@comune.bologna.it (G.B.)

**Keywords:** citizen, green infrastructure, green urban space, park, physical activity, green exercise, health status, psychological wellbeing

## Abstract

Practicing regular physical activity in green spaces has been invocated as a promising strategy for improving wellbeing in urban settings. The aim of the study was to assess the effect of a structured park-based physical activity intervention, the “Moving Parks” project, on citizens’ wellbeing at the time of COVID-19. The intervention was carried out in six public parks in Bologna (Italy) and administered by qualified instructors (from May 2021 to September 2021). The Psychological General Well Being Index short form questionnaire was administered before and after the three months of outdoor activities. A total of 328 participants completed the questionnaire at the beginning and at the end of the project. In September 2021, all psychosocial domains of the questionnaire (anxiety, depressed mood, self-control, positive well-being, vitality energy, and vitality-tiredness) significantly improved in the female sample (*p* value < 0.01) and only the last two in the male sample (*p* value < 0.05). The “Moving Parks” project seems to be able to improve citizens’ psychological wellbeing, particularly in women.

## 1. Introduction

Physical inactivity (PI) is an established risk factor for premature mortality and a global public health problem [1]. This term is used to refer to inability to achieve the recommended levels of physical activity for health [2], which for adults correspond to at least 150 min of moderate-intensity, or to 75 min of vigorous-intensity physical activity (PA) or some equivalent combination of moderate-intensity and vigorous-intensity aerobic PA, per week [3]. The World Health Organization (WHO) estimated that more than a quarter of people worldwide do not comply with these guidelines [4]. This insufficient amount of PA raises the risk for obesity, diabetes, and for other non-communicable diseases. [5] The most recent data show that the global burden attributable to PI ranges from 1.6% for hypertension to 8.1% for dementia, and it is responsible for 7.2% of all-cause deaths [6].

Physical activity instead provides many benefits for health, it plays an important role in maintaining a healthy body weight, because of the vicious circle linking obesity and PI [7]. Other benefits include bone and muscle strength promotion, lower risk for cardiovascular disease, better sleep, reduced feelings of anxiety, and boosted endurance [8,9].

Furthermore, it has been shown convincingly that increased sitting time, independently of leisure time PA, is a significant predictor of adverse health outcomes and it has been estimated that each additional hour of sitting time results in an increase of $126 in annual healthcare costs in older adults [10,11].

Unfortunately, in the effort to control the spread of the COVID-19 pandemic, many opportunities for being physically active were suspended and a declining trend of levels of PA has occurred in all age groups [12].

For this reason, Hall et al. pointed out that we are currently confronted with two pandemics occurring at the same time (COVID-19 and PI pandemic) and they predict a troubling future, where people will continue to move less and then will experience a poorer health trajectory as a result [13].

The negative effects of the lockdown have been reported not only on PA levels but also on general health in several studies [14,15] and particularly for Italy in the review by Zaccagni et al. [16]. As expected, this review highlights a significant reduction in the amount of performed PA compared to before lockdown, in both the general population and in individuals with chronic conditions [16]. This fact has negative consequences on both general health, in terms of increased body mass, and on specific chronic conditions, especially obesity and neurological diseases [17,18]

Given this strong link between PA and major non-communicable diseases, among the nine WHO global targets for improving the prevention and treatment of non-communicable diseases to be attained by 2025, one is the 10% relative reduction in prevalence of insufficient PA [19]. However, if this current trend continues, the 2025 global PA target will not be met [2], thus now more than ever there is an increasingly strong need to promote PA for the population.

Green space may promote PA and several studies have found that people living in areas with more green space undertake more PA [20]. In particular green exercise, a term referring to PA taking place in outdoor environments with exposure to nature [21], has been found to have multiple positive effects, such as sustaining physical health [22], reducing stress and anxiety [23], improving self-esteem and mood [20], and mental focus [24,25]. Thus, albeit it is evident that exercise can improve health, a growing body of evidence shows that green exercise can produce greater physical and mental health benefits than exercise in other settings [26,27].

Petrunof et al., in their study on the associations between park access, park use, park PA time and wellbeing, found that the association between the duration of PA in parks and wellbeing was particularly strong and evident, suggesting that promoting PA in parks, is a promising strategy for improving wellbeing in urban settings [28].

Many researchers have used these findings to argue the potential importance of parks and green spaces during the pandemic for maintaining or improving health [29]. Heckert and Heckert conducted a scoping review on the relationship between health and green infrastructure use (such as use of parks, natural areas, private gardens) during the COVID-19 pandemic. Based on six articles, their findings showed mixed results for COVID-19 specific health outcomes (i.e., COVID-19 reproductive rate, cases, and deaths), while improved non-COVID-19 health outcomes (i.e., depression, sleep quality, life satisfaction) were found from 15 studies, in particular regarding improved mental health [29]. 

The COVID-19 pandemic represents an unprecedented opportunity for exploring the deep integration between green infrastructure and public health [29].

The Municipality of Bologna (Italy) in collaboration with the Department of Public Health, the local sports clubs and other stakeholders launched, from 2010, the initiative “Moving parks project” to promote in spring and summer time green exercise among their citizens and potentially to improve their quality of life.

The aim of the present study was to assess whether the “Moving Parks project”, during spring and summer 2021, one year and a half since the start of the COVID-19 pandemic has increased the practice of physical activity even by people who did not practice it before and whether it has influenced the intention to continue. Another goal was to evaluate if the project may have contributed to the improvement of the psychological well-being of its participants. We hypothesize that citizen participation in park-based PA intervention affects the intention to practice PA and improves their psychological well-being.

## 2. Materials and Methods

### 2.1. The Project

The “Moving Parks project” was created by the Municipality of Bologna, a northern Italian city with approximately 391,686 inhabitants in January 2021.

The project was carried out in six public parks from May to September 2021 for a total of 90 days and involved 20 sport associations offering different types of organized activities (i.e., yoga, postural training, Nordic walking; etc.). All activities were free of charge and administered by qualified trainers.

Different strategies to enrol citizens in the project were adopted such as billposting of posters in public spaces and buses, television advertising, distribution of fliers in many local commercial activities (i.e., pharmacies, shops), newsletters, and social media. 

The study design was longitudinal with two different times of evaluation: the first at the beginning of the project (pre May 2021) and the second at the end of the project (post September 2021). Participants were directly approached by research teams in parks who also provided the administration of questionnaires. Participants were enrolled if they met these criteria: aged ≥18 years; able to write and read in Italian.

Approval for the study was obtained from the University of Bologna Bioethics Committee, (Prot. n. N 169182).

### 2.2. Questionnaire 

A structured anonymous questionnaire was used to investigate the participants’ PA habits and their psychological wellbeing before and after the three months of green exercise. This questionnaire, previously described [30], was divided into three parts: (1) general information about the participants, (2) information about their PA habits, and (3) psychological general wellbeing using the Psychological General Well Being Index short form questionnaire (PGWB-S) [31].

The PGWB-S is the short version of a 22-item HRQoL questionnaire developed in the US. 

This short version explores the following six HRQoL domains: anxiety, vitality-energy, depressed mood, self-control, positive well-being, and vitality-tiredness [32]. Each possible answer (“none of the time”, “a little bit of the time”, “some of the time”, “a good bit of the time”, “most of the time” and “all of the time” was transformed into a discrete observation to obtain only items with a Likert scale, where 1 means the poorest score and 6 means the best one. The questionnaire used during this project was validated by Toselli et al. using data from a previous study [30]. The questionnaires were administered both on paper and as an online survey using Google Moduli Form. 

### 2.3. Statistical Analysis 

Continuous variables were described using mean and standard deviation (±SD), while categorical and Bernoulli variables were described through absolute and relative frequencies. 

Normal distribution of dependent continuous variables was assessed graphically using density graphs and tested with the Shapiro–Wilk test. 

The mean comparisons between continuous variables were analyzed using the student *t*-test (*t*), whereas the mean comparisons within the same continuous variables at different times were tested by the paired student *t*-test. The proportion differences were evaluated with the Z test of proportion (Z). The significance level was set as *p* < 0.05 (*p*). 

Statistical analyses were carried out using STATA software, version 17 (StataCorp LP, College Station, TX, USA). [33]

## 3. Results

Figure 1 shows the survey flowchart. At the beginning of the project (May 2021), a cohort of 619 subjects was enrolled in the study and only 337 returned the post-questionnaire in September 2021. Of these, nine participants were excluded for missing data. Finally, a total of 328 participants were included in the analysis.

Table 1 shows the sex and age characteristics of the participants. The age distributions were divided into three categories: 18–44 years old (*n* = 153, 46.65%), 45–64 years old (*n* = 115, 35.11%), elder than 65 years (*n* = 50, 15.24%). The proportion of 25.5% of the younger participants (*n* = 39) began to practice PA with the “Moving Parks project”, whereas the proportions were 21.74% (*n* = 25) and 32% (*n* = 16) for the 45–64 and over 65 groups, respectively. In addition, 59.45% (*n* = 195) of the participants lived near the parks in which the activities were proposed. Finally, a greater proportion of female participants who attended the ‘Moving Park project’ was observed in 2021 compared to the previous decade (Z = 9.02, *p* < 0.0001), while the percentage of male participants did not change (Z = 0.23, *p* = 0.82).

In Table 2 the anthropometric characteristics, intention and importance to practice PA of the participants are reported.

Significant differences were observed in BMI and weight status between sexes both at baseline and follow-up, since women showed a higher prevalence of underweight and normal weight subjects. As a consequence, BMI is always significantly lower in women than in men. Only in females was a significant decrease in BMI observed between baseline and follow-up. Conversely, no proportion differences were observed within and between sexes in the planification of practice PA at the end of the “Moving Park project”, but male participants reported a higher mean of hours in post evaluation. In addition, both female and male samples retained the practice of PA more importantly after the project participation. 

In addition, an investigation on participants lifestyle habits was assessed in order to understand which ways participants used to travel and whether they used stairs or elevators. Female subjects preferred to walk or cycle more than males, with a significant difference in post evaluation (F = 55.29%, M = 40%; Z = 2.14, *p* < 0.05). Additionally, both sexes increased the use of cars or motorcycles for city transport after the project (M: pre = 42.16%, post = 50.67%; F: pre = 32.73%, post = 34.90%), with a significant difference in post-test (Z = −2.48, *p* = 0.01). As regards the use of stairs or elevators, a significant difference was found between sexes before the project, where males preferred the stairs compared to female subjects (Z = 2.85, *p* < 0.01). Although no significant results were found, after the project both sexes groups increased elevator usage.

Table 3 shows psychological wellbeing before and after attending the “Moving Parks” 

A significant improvement of wellbeing at the end of the project was observed in women for all the six domains. In men an improving trend was observed only in domain two and domain six. Significant differences were observed in pre- and post-evaluation between genders in domains four (“feel emotionally stable”), five (“feel cheerful”), six (“feel tired”)).

## 4. Discussion

The aim of the present study was to evaluate the effect of the “Moving Parks project”, a PA intervention conducted during the spring and summertime in 2021, one year a half since the start of the COVID-19 pandemic, on physical activity practice, on the intention to continue it, and on citizens’ psychological wellbeing. Our results appear to confirm the effectiveness of the proposal on the expected outcomes. It is to be considered that, after China, Italy was the second country in the world to be affected by the COVID-19 pandemic and the first country in Europe to implement a national lockdown to contain the spread of COVID-19. As a result, like many other activities, in Italy, gyms, pools, fitness and dance studios were closed until the end of May 2021 (only physiotherapy centers were open). In particular, sports participants stopped training due to social distancing restrictions and quarantine guidelines and on the use of sports facilities [34]. Therefore, the project was proposed at the end of the restrictions. It should be remembered that the closure of sports facilities due to COVID-19 prior to the considered period may have had a negative impact on mental health and a decrease in the level of happiness, since continuous sports practice leads to mental health happiness due to exercise [35]. Since the beginning of the SARS-CoV-2 outbreak, researchers have conducted several studies on the importance of participation in PA, also during self-quarantine, on mental health and well-being [36,37,38]. Reports state that for those who participate in physical activities such as sport for all, discover their active side and achieve new challenges and efforts on their own, which positively affect life satisfaction [39,40,41,42]. In the study period, probably people were relieved after the easing of quarantine policies; however, they could have been affected by the realistic fear of COVID-19 infection and so outdoor PA practice in green space may have given them more safety.

Most participants were women and, there was a large number of them who took part for the first time in the “Moving Parks” initiative in 2021 (65.32%) with respect to men. The willingness of the participants to follow the project is evidenced by the percentage of subjects (about one third) willing to move from their own station to follow the proposed activity of the project.

A considerable percentage of subjects (about a quarter) started to practice PA with the project, with a prevalence in females. Van Uffelen et al. reported that women preferred activities at a fixed time, whereas men were more likely to prefer activities that require skill and practice, that are vigorous and that involve competition [43]. In addition, women were more likely to prefer supervised activities, and activities with people the same age and gender as them. Thus, women have stronger preferences than men for whom they are active with, which matches with the findings showing that they are more likely to be motivated by social factors. Waters et al. (2011) found that most participants in a PA intervention trial were healthy but sedentary women (63%), and middle-aged [44]. In addition, women are more likely to respond to surveys than men [45,46]. Among those who started PA with the projects, a significant difference connected to age was observed: in females the 18–44 year old woman prevailed, while in males rather the >65 aged man. The data regarding women are in disagreement with previous studies that showed that parks were generally underutilized by young people [47].

Moreover, young people tend to spend most of their leisure time on the internet, rather than engaging in outdoor activities [43]. Probably the restrictions and relative isolation due to COVID-19 that characterized the period preceding the study fostered even in the youngest the need to be outdoors and to share activities with other people.

As regards lifestyle habits, women showed healthier habits: a significantly higher number of women than men declared using the stairs or walking, while a higher number of men than women preferred to use car, motorcycle, or scooter.

The intention to continue PA practice at the end of the project was very high. In this regard, it is important to note that consideration of participation intention precedes that of intention. Intention means that each individual’s beliefs are transferred to concrete actions towards the planned future [48]. The term, “intention” should necessarily be premised on all participatory actions of each individual. Similarly, the intention to participate in sports also indicates the decision on which situation to participate in, and the state of final resolution accordingly [49]. Researchers have reported that level of PA declined as age grew [50,51] and intrinsic motivation was identified as the only predictor of daily PA that can be measured objectively and might be positively linked to exercise adherence [52]. Indeed, participants with higher levels of intrinsic motivation have been shown to persist in activities for longer, and report higher levels of adherence [53]. In light of such considerations, the results of the present study appear to be very positive. In support of this, in both genders a significant increase in the importance given to the practice of PA regularly was observed between baseline and follow-up.

After the participation in the “Moving Parks” project, all psychosocial parameters significantly improved in women and in vitality-energy and vitality-tiredness domains in men. The improvement of these parameters is in line with the data regarding pre-COVID-19 pandemic: using data from the past project’s editions, where a reduction in the state of tension, sadness, and fatigue, and an improvement in the state of energy, serenity, and vitality of the participants was observed [30].

This study confirms that physical-activity interventions have beneficial effects on wellbeing [54,55,56] supporting its efficacy also in young adults. This represents the strength of the study, since most of the studies in literature have examined older adults and, on the other hand, there is less evidence with regard to young and midlife adults [55]. In addition, there are not many studies on this topic and in general they were not conducted in the “real world” at the time of COVID-19.

The main limitation of the study concerns the absence of a control group, which might have allowed a better interpretation of the results. The numerosity of the follow-up sample has nearly halved, probably due to the summer holidays and/or the resumption of work activities. In addition, no information on the level of education and real age were collected. The type of activities carried out during the Moving Park Project were not collected in the questionnaires. Nevertheless, it is necessary to take into account that summertime and the illusion of the end of the pandemic could have influenced the results. Further research should consider collecting additional quantitative and objective outcomes such as type of activities performed and to carry out a comparison with a control group in order to establish the effectiveness and the dose-response relations of the intervention.

Although more rigorous research is needed, our results seem to support the health-related benefits generated by PA in parks. As reported by Larson and Hipp, these benefits during the COVID-19 pandemic could be more conspicuous now than ever before [57].

## 5. Conclusions

COVID-19 has negatively affected peoples’ lifestyle habits and quality of life.

The “Moving Parks” project appears to have been able to improve citizens’ psychological wellbeing, particularly in women. Engaging regular PA in nature or green space has been indeed invocated as a promising strategy to improve wellbeing in urban settings, especially when indoor physical activity is not allowed.

Outdoor physical activity is an appropriate intervention to improve health, wellbeing, and the QoL of the population Therefore, the government, health authorities, health care professionals, and other key decision makers should consider these types of programs and initiatives that capitalize on nature-based opportunities.

Finally, the promotion of adequate health habits is essential to cope with the global health challenge of non-communicable diseases.

## Figures and Tables

**Figure 1 ijerph-19-06028-f001:**
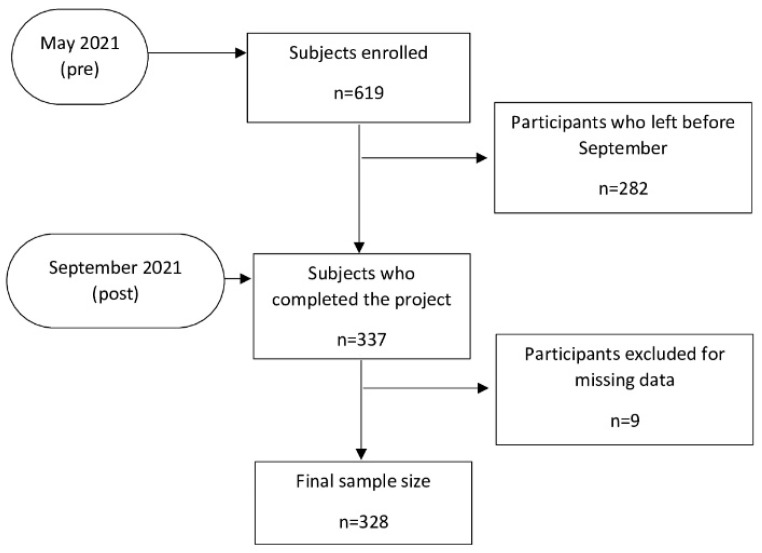
Flow chart of participants.

**Table 1 ijerph-19-06028-t001:** Proportion differences between sexes.

General	Female		Male		∆	
	*n*	%	*n*	%	Z	*p*
	253	77.13	75	22.87	23.3	<0.0001 *
Age distribution						
18–44 years	131	39.94	32	9.76	4.88	<0.0001 *
45–64 years	87	26.52	28	8.57	3.27	<0.001 *
>65 years	35	10.67	15	4.57	1.6	0.11

Note: *n*, number of observations; %, percentage; ∆, differences between sexes; Z, test of proportion Z; *p*, *p*-value; *, statistically significant.

**Table 2 ijerph-19-06028-t002:** Mean and proportion differences within and between genders.

	Females (*n* = 253)		Males (*n* = 75)		∆ Female (pre-post)		∆ Male (pre-post)		∆ Female- Male (pre)		∆ Female- Male (post)	
	Pre	Post	Pre	Post	p. t or Z	*p*	p. t or Z	*p*	t or Z	*p*	t or Z	*p*
**Anthropometric characteristics** [Mean (±SD)]												
height	162.03 (±6.1)	-	175.68 (±6.22)	-	-	-	-	-	−16.94	<0.0001 *	-	-
weight	60.35 (±9.44)	59 (±9.41)	75.93 (±11.69)	75.25 (±10.1)	1.8	0.07	0.41	0.68	−11.8	<0.0001 *	−12.9	<0.0001 *
BMI	23.05 (±4)	22.47 (±3.43)	24.64 (±3.88)	24.35 (±2.79)	2.12	<0.05 *	0.53	0.6	−3.04	<0.01 *	−4.34	<0.0001 *
Weight status [%]												
underweight	7.51	10.67	1.33	2.66	−1.63	0.1	−0.72	0,47	1.96	<0.05 *	2.14	<0.05 *
normalweight	73.52	68.77	58.67	58.66	1.63	0.1	0.01	0.99	2.47	0.01 *	1.63	0.1
overweight	15.81	17.79	34.67	34.66	−0.82	0.41	0.01	0.99	−3.58	<0.001 *	−3.12	0.001 *
obese	3.16	2.77	5.33	4.00	0.38	0.7	1.35	0.18	−0.88	0.38	−0.54	1
Plan to practice PA at the end of the project (yes) [%]	97.23	96.44	95.83	97.33	0.68	0.5	−0.81	0.42	0.62	0.54	−0.38	0.71
Frequency planned for practice (hours) [Mean (±SD)]	2.29 (±0.67)	2.22 (±0.71)	2.43 (±0.71)	2.48 (±0.67)	0.9	0.37	−0.13	0.89	−1.49	0.14	−2.7	<0.01 *
Importance to practice PA regularly [Mean (±SD)]	5.3 (±1.01)	6.05 (±0.97)	5.3 (±1.17)	6.25 (±0.92)	−36	<0.0001 *	−17.54	<0.0001 *	−0.02	0.99	−2.7	<0.01 *

Note: PA, physical activity; *n*, number of observations; SD, standard deviation; t, student’ *t*-test; p. t, paired student’ *t*-test; Z, Z-test of proportion; *p*, *p*-value; ∆, difference; *, statistically significant.

**Table 3 ijerph-19-06028-t003:** Psychological differences within and between sexes.

	Females (*n* = 253)		Males (*n* = 75)									
	Pre	Post	Pre	Post	∆ Female (pre-post)		∆ Male (pre-post)		∆ Female-Male (pre)		∆ Female-Male (post)	
	Mean (±SD)	Mean (±SD)	Mean (±SD)	Mean (±SD)	p. t	*p*	p. t	*p*	t	*p*	t	*p*
Feel nervous	4.51 (±1.33)	4.94 (±1.14)	4.81 (±1.3)	5.09 (±0.98)	−4.12	<0.0001 *	−1.5	0.11	−1.73	0.09	−1.03	0.31
Feel full of energy	4.14 (±1.01)	4.55 (±0.87)	4.41 (±1.04)	4.73 (±0.76)	−4.78	<0.0001 *	−2	0.05 *	−2.05	0.05 *	−1.7	0.09
Feel downhearted	4.42 (±0.97)	4.72 (±0.87)	4.63 (±0.93)	4.84 (±0.85)	−3.67	<0.0001 *	−1.44	0.16	−1.65	0.1	−1.09	0.29
Feel emotionally stable	3.77 (±1.17)	4.05 (±1.21)	4.12 (±1.21)	4.35 (±1.18)	−2.7	<0.01 *	−1.08	0.28	−2.27	<0.05 *	−1.92	0.05 *
Feel cheerful	3.68 (±0.99)	3.96 (±1.1)	4.04 (±1.11)	4.29 (±1.08)	−3.08	<0.01 *	−1.39	0.17	−2.68	0.01 *	−2.3	<0.05 *
Feel tired	4.23 (±0.96)	4.59 (±0.91)	4.49 (±0.84)	4.97 (±0.77)	−4.42	<0.0001 *	−3.64	<0.001 *	−2.19	<0.05 *	3.28	0.001 *

Note: *n*, number of observations; SD, standard deviation; t, student’ *t*-test; p. t, paired student’ *t*-test; *p*, *p*-value; ∆, difference; *, statistically significant.

## Data Availability

Data may be requested from authors.

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
