# Peer review of "Effect of a Park-Based Physical Activity Intervention on Psychological Wellbeing at the Time of COVID-19"

_ijerph, 2022, doi:10.3390/ijerph19106028_

Round 1
Reviewer 1 Report
Title
Overall, the title of the manuscript reads well. However, the title seems a bit too long. The Authors could consider shortening the title. For example, “results from the 3 Italian Project “Moving Parks”” might be redundant.
Abstract
Overall, the abstract is well written. There are only some minor concerns. For example, “A total of 623 and of 351…” is a bit confusing.
Authors state that significant increase was in all psychosocial parameters, but we still don’t know yet about these parameters.
Introduction
There are several short paragraphs which should be elaborated or integrated with other paragraphs. Please rework the introduction section in terms of this comment.
There are also several strong statements without any reference, for example: “Physical inactivity (PI) is a global public health problem”. I believe there is rather indirect than direct relationship, also this statement needs support by reference.
For clarity, the introduction section could end with “the present study” paragraph.
Materials and Methods
Currently, materials and methods are described too vague. Please be more specific and focus on only important and relevant information.
Results
The Authors could improve the quality of Figure 1.
Also, Table 1 does not fit very well, please reorganise this table a bit (this applies also to Table 2).
Please also check Table 3, it seems something is mixed up.
Discussion
Overall, the discussion is well written. However, I ask Authors to make more practical recommendations to the reader. Also, the Authors discuss about the intrinsic motivation in the discussion paragraph. I believe it is important to highlight that recent research has clearly demonstrated that the intrinsic motivation is the only predictor of objectively measured daily physical activity (Kalajas-Tilga et al., 2020).
Kalajas-Tilga, H., Koka, A., Hein, V., Tilga, H., & Raudsepp, L. (2020). Motivational processes in physical education and objectively measured physical activity among adolescents. Journal of Sport and Health Science, 9(5), 462–471. https://doi.org/10.1016/j.jshs.2019.06.001
Authors are also recommended to discuss more about the strengths and limitations of the current study. Also, I believe more suggestions for future research could be made by the Authors.
Author Response
Open Review
( ) I would not like to sign my review report
(x) I would like to sign my review report
English language and style
( ) Extensive editing of English language and style required
( ) Moderate English changes required
(x) English language and style are fine/minor spell check required
( ) I don't feel qualified to judge about the English language and style
Yes |
Can be improved |
Must be improved |
Not applicable |
|
Does the introduction provide sufficient background and include all relevant references? |
( ) |
(x) |
( ) |
( ) |
Are all the cited references relevant to the research? |
( ) |
(x) |
( ) |
( ) |
Is the research design appropriate? |
( ) |
(x) |
( ) |
( ) |
Are the methods adequately described? |
( ) |
(x) |
( ) |
( ) |
Are the results clearly presented? |
( ) |
( ) |
(x) |
( ) |
Are the conclusions supported by the results? |
( ) |
(x) |
( ) |
( ) |
Comments and Suggestions for Authors
Title
Overall, the title of the manuscript reads well. However, the title seems a bit too long. The Authors could consider shortening the title. For example, “results from the 3 Italian Project “Moving Parks”” might be redundant.
Authors
We would like to thank the reviewer for comment and suggestion that help us to improve our manuscript. We revised the title.
Overall, the abstract is well written. There are only some minor concerns. For example, “A total of 623 and of 351…” is a bit confusing.
Authors state that significant increase was in all psychosocial parameters, but we still don’t know yet about these parameters.
Authors
We would like to thank the reviewer for this comment. We revised the sentence as you suggested. In consideration of the psychosocial parameters, we stated in methods section that “The Psychological General Well Being Index short form questionnaire was administered before and after the three months of outdoor activities” however we revised the sentence to better explain the content.
There are several short paragraphs which should be elaborated or integrated with other paragraphs. Please rework the introduction section in terms of this comment.
There are also several strong statements without any reference, for example: “Physical inactivity (PI) is a global public health problem”. I believe there is rather indirect than direct relationship, also this statement needs support by reference.
For clarity, the introduction section could end with “the present study” paragraph.
Authors
We would like to thank the reviewer for this comment. We revised introduction section adding more references and information. (Please see Introduction section)
Materials and Methods
Currently, materials and methods are described too vague. Please be more specific and focus on only important and relevant information.
Authors
We would like to thank the reviewer for this suggestion. We revised methods section trying to be more specific and focused only on relevant information.
Results
The Authors could improve the quality of Figure 1.
Also, Table 1 does not fit very well, please reorganise this table a bit (this applies also to Table 2).
Please also check Table 3, it seems something is mixed up.
Authors
We would like to thank the reviewer for this suggestion. After performing new statistical analysis, we revised all the Tables and Figure 1.
Discussion
Overall, the discussion is well written. However, I ask Authors to make more practical recommendations to the reader. Also, the Authors discuss about the intrinsic motivation in the discussion paragraph. I believe it is important to highlight that recent research has clearly demonstrated that the intrinsic motivation is the only predictor of objectively measured daily physical activity (Kalajas-Tilga et al., 2020).
Authors
We would like tot to thank the reviewer for this comment. We added a sentence about further research, please see the discussion section. We agree with the reviewer about the intrinsic motivation. We added the suggested reference to strengthen our sentence.
Kalajas-Tilga, H., Koka, A., Hein, V., Tilga, H., & Raudsepp, L. (2020). Motivational processes in physical education and objectively measured physical activity among adolescents. Journal of Sport and Health Science, 9(5), 462–471. https://doi.org/10.1016/j.jshs.2019.06.001
Authors are also recommended to discuss more about the strengths and limitations of the current study. Also, I believe more suggestions for future research could be made by the Authors.
Authors
We would like to thank the reviewer for this suggestion. We already included the strength of our study and we revised the limitation section.
Submission Date
19 April 2022
Date of this review
21 Apr 2022 10:52:33

Reviewer 2 Report
The analysis of the physical activity in green spaces is very old, where in a brief survey we were able to find studies dating from 1708. The study is interesting but some factors are missing to clarify the reader. My assessment will be global in some points and specify in others.
In the abstract, Why does the age of the participants not appear? In the abstract, just put the final n!
What are the global results, can be here in the summary and not only the indication of significant changes, being able to present the total p-value of the questionnaire
In the introduction, care should be taken with the formatting of the text, (E.g. adults [8,9]) and there should be space (adults [8, 9]), or not. During the text Sometimes your paragraphs have previous space, sometimes they don't, there must be coherence in this. Review the entire text.
Attention to English, sex is not written, but gender.
The methodology is well described, but I call attention to the bibliographic reference referring to the statistical data because it does not show where it was published.
In the Materials and Methods Why not just an intervention methodology? Was it analyzed how the activity typology may have influenced the results?
In your results it was not 956! The same participant is not counted twice. There were 619 at the beginning and 282 left/left the study, leaving the final number with 337.
There were then 337 valid participants! Who did the pre and post test, right?
This number is not the one shown in the abstract! (623, 351)?
Something that raised some concern was that the data presented did not correspond correctly between its abstract and its methodology. As it is not clear how it reached these people, what are the inclusion and exclusion criteria of your study?
In table 1, this n of the Age distribution factor of who started PA with the project (513 Woman and 107 Man), makes no sense, since it indicated that only 157 (135 Woman, and 22 Man) started the PA with de project.
In table 2, this type of anthropometric comparison only makes sense if evaluating the same sample number, who started and ended the study, the rest dropped out, and should not be counted in the final evaluation, only in the presentation of the results.
In table 2, this n (774 females and 182 males) is not the sum, because the 262 females and 75 males in the Post were in Pre, so the same person is not counted twice in the study. The height has changed because the n has changed, there is no point in the statistical analysis of the difference here, only if it is between men and women.
The reliability of your results is only valid if you compare the same n in the pre and post test. Because the statistical difference may come from who dropped out of the study. What biases your results
I realize how hard it is to carry out such a study, but I recommend a review before it is approved.
Author Response
Open Review
(x) I would not like to sign my review report
( ) I would like to sign my review report
English language and style
( ) Extensive editing of English language and style required
( ) Moderate English changes required
(x) English language and style are fine/minor spell check required
( ) I don't feel qualified to judge about the English language and style
Yes |
Can be improved |
Must be improved |
Not applicable |
|
Does the introduction provide sufficient background and include all relevant references? |
(x) |
( ) |
( ) |
( ) |
Are all the cited references relevant to the research? |
(x) |
( ) |
( ) |
( ) |
Is the research design appropriate? |
(x) |
( ) |
( ) |
( ) |
Are the methods adequately described? |
(x) |
( ) |
( ) |
( ) |
Are the results clearly presented? |
( ) |
( ) |
(x) |
( ) |
Are the conclusions supported by the results? |
( ) |
(x) |
( ) |
( ) |
Comments and Suggestions for Authors
The analysis of the physical activity in green spaces is very old, where in a brief survey we were able to find studies dating from 1708. The study is interesting but some factors are missing to clarify the reader. My assessment will be global in some points and specify in others.
Authors
Would like to thank you the reviewer for the comment and suggestion that help us to improve our work.
In the abstract, why does the age of the participants not appear? In the abstract, just put the final n!
Authors
We would like to thank the reviewer for the comment. The sample was heterogeneous and within the questionnaires we collected the age-range of the participants (es. <18, 18-44, 45-64, >64) as shown in Table 1. We made this methodological decision in order to respect privacy policy. We included a statement in limitation section and we specified in the abstract that participants are adults.
What are the global results, can be here in the summary and not only the indication of significant changes, being able to present the total p-value of the questionnaire.
Authors
Following reviewer’s suggestion, we revised abstract results section to clarify.
In the introduction, care should be taken with the formatting of the text, (E.g. adults [8,9]) and there should be space (adults [8, 9]), or not. During the text Sometimes your paragraphs have previous space, sometimes they don't, there must be coherence in this. Review the entire text.
Attention to English, sex is not written, but gender.
Authors
We would like to apologize for these mistakes. We revised introduction section in order to be more coherent. We agree that gender is a grammatically appropriate than sex, however in clinical and Health application they have two different meanings. We ask to participant’s information on sex with no explanation on how gender differences could impact and affect their answer questionnaire related. However, to better understand our opinion we report a recent study that investigate on these differences: https://doi.org/10.1038/embor.2012.87
The methodology is well described, but I call attention to the bibliographic reference referring to the statistical data because it does not show where it was published.
Authors
We would like to thank the reviewer for appreciating our methods section. We added the requested information.
In the Materials and Methods Why not just an intervention methodology? Was it analyzed how the activity typology may have influenced the results?
Authors
We agree with the reviewer's comment, knowing the type of activity would have enriched the results. Unfortunately, the activities proposed in this project are very numerous and it was not possible to categorize them. We added this limitation also in the discussion section and in suggestion for future research.
In your results it was not 956! The same participant is not counted twice. There were 619 at the beginning and 282 left/left the study, leaving the final number with 337.
Authors
We would like to apologize for these mistakes. We corrected the numbers of participants also revising all Tables and Figure.
There were then 337 valid participants! Who did the pre and post test, right?
Authors
We would like to thank the reviewer for this comment. In these past weeks we collaborated with the associations, and we obtained a code useful to match the pre-post test of each involved participants respecting anonymous methodology. Then, we used the paired t-test to compare pre and post-continuous variables.
This number is not the one shown in the abstract! (623, 351)?
Authors
We would like to apologize for these mistakes. We corrected the number in the abstract.
Something that raised some concern was that the data presented did not correspond correctly between its abstract and its methodology. As it is not clear how it reached these people, what are the inclusion and exclusion criteria of your study?
Authors
We would like to thank the reviewer for this comment. All the people who participated in the project and who voluntarily gave their consent to fill the pre- and post-questionnaires were enrolled in the study. We clarify this aspect in the text (Material and Methods section)
In table 1, this n of the Age distribution factor of who started PA with the project (513 Woman and 107 Man), makes no sense, since it indicated that only 157 (135 Woman, and 22 Man) started the PA with de project.
In table 2, this type of anthropometric comparison only makes sense if evaluating the same sample number, who started and ended the study, the rest dropped out, and should not be counted in the final evaluation, only in the presentation of the results.
In table 2, this n (774 females and 182 males) is not the sum, because the 262 females and 75 males in the Post were in Pre, so the same person is not counted twice in the study. The height has changed because the n has changed, there is no point in the statistical analysis of the difference here, only if it is between men and women.
Authors
We would like to thank the reviewer for this comment. Following reviewer’s suggestion, we revised the statistical analysis focused only on the number of participants that started and ended the project.
The reliability of your results is only valid if you compare the same n in the pre and post test. Because the statistical difference may come from who dropped out of the study. What biases your results
I realize how hard it is to carry out such a study, but I recommend a review before it is approved.
Authors
We would like to thank the reviewer for this final comment. After careful revised our work, we agreed with the reviewer that it was essential to work with paired data. For this reason, we matched data using codes and made the statistical analyzes again based on participants that started and ended the project.
Submission Date
19 April 2022
Date of this review
21 Apr 2022 16:26:13

Round 2
Reviewer 1 Report
Authors have done well job on revising the manuscript.
Author Response
Open Review
( ) I would not like to sign my review report
(x) I would like to sign my review report
English language and style
( ) Extensive editing of English language and style required
( ) Moderate English changes required
(x) English language and style are fine/minor spell check required
( ) I don't feel qualified to judge about the English language and style
Yes |
Can be improved |
Must be improved |
Not applicable |
|
Does the introduction provide sufficient background and include all relevant references? |
(x) |
( ) |
( ) |
( ) |
Are all the cited references relevant to the research? |
(x) |
( ) |
( ) |
( ) |
Is the research design appropriate? |
(x) |
( ) |
( ) |
( ) |
Are the methods adequately described? |
(x) |
( ) |
( ) |
( ) |
Are the results clearly presented? |
(x) |
( ) |
( ) |
( ) |
Are the conclusions supported by the results? |
(x) |
( ) |
( ) |
( ) |
Comments and Suggestions for Authors
Authors have done well job on revising the manuscript.
Authors
We would like to thank the reviewer for comments and suggestions that allow us to improve our manuscript.
Submission Date
19 April 2022
Date of this review
10 May 2022 09:47:39

Reviewer 2 Report
Dear authors,
Thank you for the answers and the improvements made to your work. These allow, in my view, a better and clearer approach to the topic and better conclusions to the results.
I hope you always keep in mind that our role is to propose improvements. In this sense and in order to fill in some details that still need a closer "look", I propose that the authors reread the entire article with the changes made to understand the logic of the chain of ideas.
I propose in the abstract the placement of the main results and the real p-value and not simply p<0.05.
I appreciate the improvements in the introduction and methodology, but I would like to point out that during the text sometimes your paragraphs have previous space, sometimes they don't, there must be coherence in this. Review the entire text.
In the results, to clarify the reader, usually a paragraph with the main details of the results is presented before a figure or table.
I congratulate you for the work, for the good discussion and coherent conclusion.
I hope to have helped to improve it,
I wish success
GD
Author Response
Open Review
( ) I would not like to sign my review report
(x) I would like to sign my review report
English language and style
( ) Extensive editing of English language and style required
( ) Moderate English changes required
( ) English language and style are fine/minor spell check required
(x) I don't feel qualified to judge about the English language and style
Yes |
Can be improved |
Must be improved |
Not applicable |
|
Does the introduction provide sufficient background and include all relevant references? |
(x) |
( ) |
( ) |
( ) |
Are all the cited references relevant to the research? |
(x) |
( ) |
( ) |
( ) |
Is the research design appropriate? |
(x) |
( ) |
( ) |
( ) |
Are the methods adequately described? |
(x) |
( ) |
( ) |
( ) |
Are the results clearly presented? |
( ) |
(x) |
( ) |
( ) |
Are the conclusions supported by the results? |
(x) |
( ) |
( ) |
( ) |
Comments and Suggestions for Authors
Dear authors,
Thank you for the answers and the improvements made to your work. These allow, in my view, a better and clearer approach to the topic and better conclusions to the results.
I hope you always keep in mind that our role is to propose improvements. In this sense and in order to fill in some details that still need a closer "look", I propose that the authors reread the entire article with the changes made to understand the logic of the chain of ideas.
Authors
We would like to thank the reviewer for helping ous to improve our work with precius comment and suggestion. We read again our work revising some part to allow a better flow during the reading.
I propose in the abstract the placement of the main results and the real p-value and not simply p<0.05.
Authors
We would like to thank the reviewer for this comment. Taking into account the word counts in the abstract it is not possible to insert all the results of the six domains. However, we have added one more p value.
I appreciate the improvements in the introduction and methodology, but I would like to point out that during the text sometimes your paragraphs have previous space, sometimes they don't, there must be coherence in this. Review the entire text.
Authors
Following reviewer’s suggestion we revised the entire text.
In the results, to clarify the reader, usually a paragraph with the main details of the results is presented before a figure or table.
Authors
We would like to thank the reviewer for this suggestion we revised results section putting the main detail of the results before figures and tables.
I congratulate you for the work, for the good discussion and coherent conclusion.
I hope to have helped to improve it,
I wish success
GD
Authors
We would like to thank the reviewer for appreciating our work and to give us fundaental advice to improve our manuscript.
Submission Date
19 April 2022
Date of this review
10 May 2022 15:38:05
